# Dietary Intakes of Folate, Vitamin D and Iodine during the First Trimester of Pregnancy and the Association between Supplement Use and Demographic Characteristics amongst White Caucasian Women Living with Obesity in the UK

**DOI:** 10.3390/nu14235135

**Published:** 2022-12-02

**Authors:** Kathy M. Redfern, Heidi J. Hollands, C. Ross Welch, Jonathan H. Pinkney, Gail A. Rees

**Affiliations:** 1Faculty of Health, University of Plymouth, Drake Circus, Plymouth PL4 8AA, UK; 2Department of Fetal Medicine, University Hospitals Plymouth National Health Service Trust, Plymouth PL6 8DH, UK

**Keywords:** pregnancy, maternal obesity, supplementation, folic acid, vitamin D, iodine

## Abstract

Folate, vitamin D and iodine are key micronutrients in pregnancy, with deficiency associated with poor maternal and infant outcomes. For folate and vitamin D especially, deficiency is more common amongst women with obesity and recommended intakes and guidance on supplementation varies worldwide. The present study aims to investigate dietary and supplementary intakes of these micronutrients amongst a population of pregnant women with obesity in the United Kingdom, alongside key maternal demographic characteristics. Expectant women (*n* = 75) with a body mass index ≥ 30 kg/m^2^ at first antenatal appointment were recruited at 12 weeks gestation. Participants were asked about their supplement use preconception and during trimester one in a baseline questionnaire which also asked about demographic characteristics. Women also completed a four day diet diary from which dietary and supplemental intakes of micronutrients intakes were estimated. Folic acid was taken by 96% of women at any point in trimester 1, whilst only 26% of women took the higher 5 mg dose recommended for women with obesity in the UK. For vitamin D and iodine, 56% and 44% of women met the UK RNI, respectively. Maternal age was positively associated with taking supplements of any kind and the 5 mg folic acid supplement, whilst parity was inversely associated with both outcomes. This study strengthens the rationale for further work to be done raising awareness of the need for women with obesity to supplement both with a higher dose of folic acid and vitamin D and to be aware of the role of iodine during pregnancy.

## 1. Introduction

Nutritional status prior to and during pregnancy influences growth and development of the fetus and general maternal health [1]. There is significant interest in the role of maternal under- or over-nutrition on outcomes such as gestational weight gain, infant birth size and other adverse pregnancy outcomes such as gestational diabetes mellitus, pre-eclampsia and pre-term delivery.

The incidence of maternal obesity is increasing worldwide, across Europe and in the United Kingdom (UK; [2,3]). Much of the literature examining diet during pregnancy amongst women with obesity is focused on energy intake, macronutrient intakes and dietary patterns, and studies examining micronutrient intakes during pregnancy have tended to focus on under-nourished women, rather than those with obesity. Women with obesity are often considered to be ‘over-nourished’ however, a recent observational study conducted in England reported that intakes of iron, vitamin D, iodine and folate were below the reference nutrient intake (RNI) for the majority of women with obesity [4].

Folate and folic acid have been the focus of much micronutrient research in pregnancy due to their role in the prevention of neural tube defects (NTDs). The neural tube closes within 4 weeks of conception, and thus, the Department of Health in the UK recommend that women supplement with 400 µg folic acid daily from pre-conception to 12 weeks gestation [5]. Supplementation has been shown to reduce the risk of NTDs in randomised controlled trials [6]. However, maternal obesity has been shown to be associated with increased risk of NTDs [7] and in the US, despite implementation of the 1998 US folate fortification program of cereal products, increased maternal BMI was associated with lower serum folate status [8]. In the UK, the Royal College of Obstetricians and Gynaecologists (RCOG) recommend that women with obesity, defined as a body mass index (BMI) ≥ 30 kg/m^2^, intending to become pregnant or already pregnant should take a higher dose of 5 mg folic acid daily until the end of the first trimester of pregnancy [9], which in the UK, is only available on prescription. A nested cohort study conducted in Dublin, Ireland, observed that women with obesity were significantly less likely to take a pre-pregnancy folic acid supplement than women with a healthy BMI and none reported taking the higher 5 mg dose, although dose was recorded for only 36% of the sample [10].

In addition to folate, vitamin D and iodine are important micronutrients during pregnancy. Iodine has long been known to be important for fetal brain development during pregnancy and a focus for the World Health Organisation (WHO) amongst women of childbearing age in developing countries [11]. However, recent data from the UK has suggested that mild to moderate iodine deficiency exists amongst women of childbearing age in the UK [12]. Recent findings from the Avon Longitudinal Study of Parents and Children (ALSPAC) suggest that mild to moderate iodine deficiency during early pregnancy is associated with impaired cognitive function in offspring [13]. Supplementation with iodine is not currently recommended for women in the UK during pregnancy, although the RNI for the general population of 140 µg/day [14] is present in the majority of pregnancy multivitamin products marketed to women in pregnancy. This is in contrast to advice from WHO/UNICEF who recommend iodine intake is increased from 150 µg to 250 µg/day in pregnancy [15], whilst the European Food Safety Authority (EFSA) recommend 200 µg/day [16]. The UK RNI may therefore be insufficient for pregnant women with a healthy BMI, and it is not known whether women with obesity have higher requirements.

The importance of adequate vitamin D status during pregnancy is important to protect fetal skeletal development and it is widely accepted that maternal vitamin D deficiency should be prevented [17]. In the UK, SACN estimate intakes of 10 µg/day will enable 97.5% of individuals, including pregnant women, to meet or exceed the target of 25-hydroxyvitamin D [25(OH)D] concentrations of 25 nmol/litre [18], whilst the WHO and EFSA recommend 5 µg/day and 15 µg/day, respectively [19,20]. In addition, many European countries at more southerly latitudes than the UK, where sunlight is likely of sufficient strength to trigger the conversion 7-dehydrocholesterol in the skin to cholecalciferol for more months of the year, have higher recommended intakes of vitamin D than the UK of 15–20 µg/day during pregnancy [21]. Obesity is a risk factor for vitamin D deficiency, which may be related to sequestering of vitamin D3 in adipose tissue, and in pregnancy, maternal obesity has been shown to increase the odds of both maternal and neonatal vitamin D deficiency [22]. Vitamin D deficiency has also been shown to be an independent risk factor for pre-eclampsia, a condition which is more frequently observed amongst women with obesity than women with a healthy BMI [23] with supplementation with vitamin D shown to reduce the risk of the reoccurrence of pre-eclampsia [24].

An observational study, conducted in Plymouth, UK, concerned the collection of weight gain, diet, physical activity, sleep and infant data amongst a cohort of pregnant women with obesity. The aim of the present study was to examine the dietary and supplementary intakes of key nutrients of interest in this population: folate, iodine and vitamin D and to investigate for any association between supplement use and key maternal demographic characteristics. We hope that our findings will be of use to health professionals involved in the dietary counselling of women pre-conception and in early pregnancy, and that we will highlight groups of women for whom interventions promoting supplementation may have the most impact.

## 2. Materials and Methods

Women aged between 18 and 40 years, with a BMI ≥ 30 and <40 kg/m^2^ at first hospital booking appointment and pregnant with a singleton pregnancy were eligible to take part in the study. Women meeting inclusion criteria were identified from their antenatal booking notes by a Research Midwife and approached by the researcher at their 12 week dating scan at Derriford Hospital in Plymouth, UK between January 2015 and December 2017. Ethical approval was obtained from the NHS Health Research Authority National Research Ethics Service and local Research and Development (R&D) approval was obtained from University Hospitals Plymouth NHS Trust.

Following recruitment, participants were visited by the researcher who obtained both verbal and written informed consent. The first visit occurred between 12 and 14 weeks gestation at which point participants answered a baseline questionnaire which asked about demographic characteristics as well as preconception supplementation habits. Participants were also given a food diary and asked to record, in as much detail as possible, all food and beverages consumed within a 4 day period following each study visit, giving details about their portion sizes using weights, household measurements, packet sizes and photographs. The 4 day period was chosen in an attempt to maximise compliance with this aspect of data collection, and it was also the same 4 day period that participants were asked to wear an accelerometer for the collection of physical activity data. The 4 day diet diary was adapted from the previously validated UK National Diet and Nutrition Survey in order to maximise validity and reliability of dietary assessment [25]. Subjects were also asked to record any dietary supplements, whether prescribed or self-bought. At the end of the 4-day period, the researcher visited the participant to collect the diary and to clarify portion sizes, the types of foods eaten and supplements taken. The researcher also asked the participant to report whether their dietary intake had been affected by complications such as pregnancy sickness or hyperemesis gravidarum. Dietary assessment data was analysed using DietPlan 7 (Forestfield Software Ltd. 2010, Horsham, UK) to generate nutritional intake data for each participant using data from UK Food Composition Tables [26]. Dietary intakes and supplemental intakes of micronutrients were estimated and reported separately. Food portion sizes were estimated from the photographs, weights given and household measurements using ‘Food Portion Sizes’ published by the Food Standards Agency in the UK [27]. When foods were missing from the database, nutrient data was obtained from the manufacturer where possible and added manually to the database. For some foods this was not possible, in which case the researcher chose a food with similar nutrient composition from the database.

## 3. Results

Of the original sample of 75 women, 66 completed at least three days of their diet diary at the end of trimester 1. Table 1 shows descriptive data for these women. Compliance with the diet diary element of the study decreased as pregnancy continued, as did the proportion of women taking a supplement of any kind from decreased from 65% of women at the end of trimester 1, to 46.6% and 46.2% at the end of trimesters 2 and 3, respectively.

Of the 66 women completing a trimester 1 diet diary, 30 (46%) reported taking a supplement containing folic acid pre-conception, while 62 (96%) reported taking a supplement containing folic acid at some point during trimester 1. Table 2 shows the dietary and supplementary intakes of folate/folic acid, vitamin D and iodine reported in diet diaries at the end of trimester 1. At this point, 42 women (64%) report taking a supplement containing folic acid, and all of these women were meeting the UK RNI for folate of 300 µg/day with the addition of supplements. Of the 24 women not reporting supplementation with folic acid at the end of trimester 1, only one woman was achieving the RNI of 300 µg through their dietary intake. There was no statistically significant difference in the dietary intakes of folate between women supplementing (221.5 ± 102.4 µg) and those not taking a supplement (182.5 ± 81.0 µg, *p* = 0.114).

A total of 37 (56%) and 28 women (42%) reported use of a supplement containing vitamin D and iodine, respectively. With the use of the supplement, these women achieved the UK RNI for vitamin D of 10 µg/day and for iodine of 140 µg/day, although when looking at the EFSA iodine recommendations of 200 µg/day, 4 women who were taking a supplement containing iodine did not meet this target. None of the women not taking a supplement achieved the RNI for vitamin D, while just 6 women (16%) achieved the UK RNI for iodine, which reduced to just 4 women (14%) when considering the EFSA recommendation. Dietary intakes of vitamin D were significantly greater amongst women who were also supplementing (2.0 ± 1.1 µg) than amongst women who did not take a supplement (1.3 ± 0.9 µg, *p* = 0.007). For women taking a supplement containing iodine, dietary intakes were higher (107.4 ± 52.2 µg) than intakes of women not supplementing (92.5 ± 55.9 µg), however, this trend did not reach significance (*p* = 0.082).

A binomial logistic regression was performed to ascertain the effects of maternal age, index of multiple deprivation, booking BMI and previous pregnancies on the likelihood of taking supplements, of any kind, in trimester 1. The logistic regression model was statistically significant, χ^2^(4) = 16.638, *p* = 0.002. Of the four predictor variables, increasing maternal age was positively and significantly associated with supplement use, while previous number of pregnancies was inversely and significantly associated with supplement use.

The RCOG recommend that women with obesity supplement with a higher 5 mg dose of folic acid during pregnancy. Table 3 shows that just 17 women (26%) were taking this higher dose at the end of trimester 1, while a further 24 were taking a supplement containing 400 µg. One woman took a supplement containing 25 µg of vitamin D although this particular participant did not consume a folic acid supplement at any point. The remaining women supplementing with vitamin D consumed a supplement containing 10 µg.

A second regression analysis was performed to ascertain the effects of the same variables previously examined for supplement use, on the likelihood of taking the recommended 5 mg dose of folic acid. Once again, the model was statistically significant χ^2^(4) = 16.488, *p* = 0.002 with maternal age and BMI positively associated with use of the higher 5 mg folic acid dose, and previous number of pregnancies inversely associated. There was no association observed between Multiple Index of Deprivation and higher 5 mg folic acid dose.

## 4. Discussion

Findings from this observational study demonstrate that whilst 46% and 96% of pregnant women with obesity took a folic acid supplement pre-conception and in the first trimester, respectively, only 26% of women took the higher 5 mg dose recommended by the RCOG. For vitamin D and iodine, 56% and 52% of women met the UK RNI, respectively. For women who did not supplement with these two micronutrients, no women met the RNI for vitamin D and only 16% of women met the RNI for iodine. Maternal age was positively associated with taking supplements of any kind and taking the 5 mg folic acid supplement, whilst parity was inversely associated with both outcomes. The present study is unique in that it focuses on supplementation trends for three key micronutrients of concern amongst pregnant women with obesity.

In terms of folic acid supplementation with the UK recommendation of 400 µg per day, our findings are in keeping with those observed amongst the general pregnant population in the UK. A total of 46% of women in our study supplemented with folic acid pre-conception, which is similar to the rate of 39% observed amongst women with a BMI > 30 kg/m^2^ who were actively planning a pregnancy in a prospective cohort conducted in women in the UK [28]. A limitation of the current study is that women were not asked whether their pregnancy was planned or unplanned, although previous studies have suggested higher rates of unplanned pregnancies in women with obesity, due to hormonal contraception failure [29], and unsurprisingly, women planning a pregnancy are more likely to be taking folic acid preconception [30].

The observation in the present study that 96% of women reported taking folic acid at any point in trimester 1 seems to be slightly higher than values reported in the literature from other studies in the UK conducted amongst women of all weights which ranged from 67–85% in three large cohort studies [31,32,33,34], but is similar to the rate observed by Cawley et al. of 96.1%, also amongst women of all weights [35]. Of these, 26% of women in the present study were taking the recommended 5 mg folic for women with obesity. Unfortunately, it is not known what proportion of these women took the 5 mg pre-conception, or whether they were prescribed the 5 mg dose once they’d engaged with a healthcare professional during their pregnancy. Therefore, it is not appropriate to compare the findings of the current study directly against those of a recent Irish study, which specifically asked women about pre-conception high dose folic acid compliance and found that no women reported taking this higher dose [10]. Similarly, Cawley et al. [35] report that just 2 of 106 women with obesity reported taking the high dose supplement in their observational study at any time in pregnancy [35].

However, despite a high proportion of women reporting taking a folic acid supplement at any point of trimester 1, by the end of trimester 1, when four-day food records were collected, just 64% of women reported taking a supplement containing folic acid. Of the 24 women not taking a supplement, only one woman met the pregnancy RNI of 300 µg through dietary intakes, leaving 42% of women in the total population with intakes below the RNI. This is concerning as although folate is important preconception for the prevention of NTDs, folate is also essential throughout the rest of pregnancy for the prevention of complications and poor birth outcomes such as anaemia, preterm birth, low birth weight and congenital heart diseases [36]. Although the UK does not recommend to women that they continue a 400 µg/day supplement beyond the end of the first trimester, it is important that women are aware of the need to consume foods rich in folate or fortified with folic acid to meet the RNI in the remainder of their pregnancy, as per NICE guidance [5].

In total, 56% of women reported supplementing with vitamin D at the end of trimester 1, all of whom met the RNI of 10 µg/day. This is considerably higher than the 27% of women meeting vitamin D recommendations in a Finnish observational study conducted amongst women with obesity [37]. Conversely, women who did not report taking a supplement containing vitamin D had mean intakes of 1.4 µg/day, and none of these women met the RNI for vitamin D. These observations are in keeping with those from a recent Irish study, in which only 1% of pregnant women were meeting the RNI for vitamin D from diet alone [38]. These findings suggest that not all women are aware of the recommendation to supplement with 10 µg vitamin D throughout pregnancy, which should be a key feature in public health nutrition campaigns, particularly those aimed at women with obesity, which is a risk factor for maternal and neonatal vitamin D deficiency [22] which in turn is a risk factor for pre-eclampsia [23].

Similarly, when examining iodine intakes in the present study, the 42% of women who reported taking an iodine containing supplement met the UK RNI of 140 µg for iodine. These observations are slightly higher than those observed amongst pregnant women in the US where only 17.8% of women reported taking a supplement containing iodine [39], despite the fact that there is a pregnancy increment in the recommended daily allowance from 150 µg to 220 µg in the USA as per Institute of Medicine recommendations [40] and that the American Thyroid Association recommend a 150 µg/day supplement [41]. Interestingly, although perhaps unsurprisingly, when considering the EFSA iodine recommendations of 200 µg/day in pregnancy, the proportion of supplementing and non-supplementing women meeting the EFSA RNI in the present study decreased compared with the lower UK RNI for which there is not a pregnancy increment. However, it is perhaps not useful to compare iodine intakes in pregnancy to those in other countries, particularly those in which iodised salt is routinely available. For example, iodised salt in the USA and Canada provides approximately 45 µg iodine per gram of salt, whereas in the UK and many other European countries, salt is not iodised. Globally, UNICEF estimate that 89% of the population are consuming iodised salt [42] but it is worth noting that although table salt may be iodised, in many countries the salt added to processed foods, which makes up a large proportion of dietary salt intake, is not iodised. In addition, despite progress towards reducing iodine deficiency globally, there are countries in Northern Europe with iodised salt programmes that are still considered iodine deficient, including Germany, Finland and Norway [43], and a recent study in South Australia suggests that even with mandatory fortification of bread with iodine contributing to iodine sufficiency, it is difficult to achieve urinary iodine concentrations >150 µg/L without additional iodine supplementation [44].

Of the women not supplementing with iodine in the present study, only 6 women (16%) met the RNI from dietary intakes, which gives possible cause for concern as iodine supplements are not currently recommended in the UK, and unlike vitamin D and folic acid, iodine is not included in the NHS Healthy Start vitamins that are available free of charge to low-income women in the UK [45]. The majority of other branded multivitamin products marketed to pregnant women in the UK that are sold in supermarkets and high street pharmacies do contain iodine.

Additionally, it is also well documented that pregnant women in the UK are generally iodine insufficient [12] despite a lack of large good quality studies [46]. In addition, studies have shown even marginal iodine deficiency in pregnancy is associated with impaired cognitive outcomes for offspring [12,47], highlighting the importance of the mineral in future preconception and pregnancy research.

Increasing maternal age was positively associated with supplement use of any kind and taking the 5 mg folic acid dose in the present study, which is in agreement with findings from other studies conducted in the UK [31], Finland [48] and USA [39], and suggests younger women in particular should be targeted for interventions aiming to increase supplement use. In addition, parity was negatively associated with supplement use and taking high dose folic acid, suggesting that women may benefit from a reminder to restart supplement use following a pregnancy when they intend to become pregnant again. It is also possible that some women may not have needed to take the higher dose of folic acid in a previous pregnancy due to their weight being lower, but it is well documented than women are more likely to start subsequent pregnancies at a higher weight and BMI than a previous pregnancy [49]. Therefore, as well as following NICE guidance to achieve a healthy weight between pregnancies [50], women also need to be made aware of the need to take a higher dose of folic acid if they plan to enter a subsequent pregnancy at a higher body weight.

The present study did not observe any associations between supplement use and deprivation, which is in contrast to many studies conducted amongst women of all weights. For example, Brough et al. report that women from higher social groups in the UK were more likely to take a folic acid supplement at any stage of their pregnancy [51]. Similarly, Alwyn et al. report that pregnant women taking a supplement of any kind during pregnancy were less likely to be living in an area with an IMD score in the lowest quartile [31]. It is worth noting that all women in the present study lived in or near Plymouth, in the United Kingdom which as a Local Authority district, has an IMD score in decile 2, placing within the 20% most deprived local authority districts in the country [52]. Although there is variation within the city, this may explain why no association was observed between IMD score and supplementation, as all women live within a city with high deprivation.

A limitation of this study is that women were all of white Caucasian origin, so we were not able to examine whether race or ethnicity predict supplement use in women with obesity. A previous meta-analysis conducted in the United Kingdom demonstrated higher levels of peri-conceptional folic acid use amongst Caucasian women when compared to women of other ethnicities [53]. Future studies should focus on examining supplementation trends for key pregnancy micronutrients in more diverse populations. Although we were able to report parity in the present study, there was no data available for women who may have suffered previous miscarriages, nor did we have information about concomitant medications which may have influenced women taking supplements in early pregnancy. In addition, we acknowledge that based on the mean reported energy intake of 1766 ± 442.6 kcal/day, it is possible and likely that some women under-reported their dietary intake, which may have led to under-estimations of dietary intakes of micronutrients. However, as this data was collected in the first trimester of pregnancy where pregnancy sickness and changes to appetite are common, we have not excluded any women from analysis.

A strength of this study was that information on dietary intakes and supplement use was recorded using a structured four-day food diary which was distributed, checked and analysed by a single nutrition researcher at each study visit and allowed the researcher to check food and supplement intakes with the women. This reduced the risks of respondent error, recall bias or inter-observer variation. The study is also the first, to the authors’ knowledge that specifically examines dietary and supplemental intakes of folate, vitamin D and iodine amongst women with obesity in the UK, with particular focus on preconception intake of the higher folic acid dose and demographic characteristics.

## 5. Conclusions

In conclusion, findings from the present study suggest that women with obesity in the UK who do not take a pregnancy micronutrient supplement are unlikely to be meeting the RNI for folic acid, vitamin D and iodine, three important micronutrients during pregnancy. Younger women and women who had been pregnant before were less likely to take any micronutrient supplement so it may be particularly important to target interventions and public health information for these women. Particular attention should be paid to folic acid supplementation and ensuring that women with obesity are made aware of the need to take a higher 5 mg dose preconception, alongside 10 µg vitamin D, as advised by the NHS and the RCOG. In the UK, the 5 mg folic acid supplement is only available on prescription, so it is imperative that GPs and midwives are aware of the higher dose recommendations for women with obesity and counsel women prior to conception if possible. This is particularly important for low-income women who may be eligible for free NHS Healthy Start vitamins, which contain the 400 µg folic acid rather than the higher 5 mg dose. Further research is required to investigate iodine status and whether there is potential for iodine supplementation in pregnancy to improve pregnancy and infant outcomes.

## Figures and Tables

**Table 1 nutrients-14-05135-t001:** Maternal descriptive data, *n* = 66.

	Mean ± SD (Range)
Maternal age, years	30.1 ± 4.5 (20.0–39.0)
Previous pregnancies, number	0.9 ± 1.0 (0.0–4.0)
Index of Multiple Deprivation, decile	4.3 ± 2.8 (1.0–10.0)
Ethnicity, white Caucasian, *n* (%)	66 (100)
Pregnancy sickness experienced, *n* (%)	53 (80)
Appetite affected due to pregnancy sickness, *n* (%)	52 (79)
Booking BMI, kg/m^2^	33.0 ± 1.9 (30.0–37.6)
Energy intake, kcal/day	1766 ± 442.6 (769–2893)
Protein intake, g/day	67.7 ± 20.7 (24.5–144.8)
Fat intake, g/day	67.9 ± 21.1 (24.6–140.4)
Carbohydrate intake, g/day	235.3 ± 64.3 (105.9–408.4)
Women taking a supplement of any kind, trimester 1, *n* (%)	43 (65.2)
Women taking a supplement of any kind, trimester 2, *n* (%) *	27 (46.6)
Women taking a supplement of any kind, trimester 3, *n* (%) **	24 (46.2)

* *n* = 58; ** *n* = 52.

**Table 2 nutrients-14-05135-t002:** Trimester 1 intakes of folate, vitamin D and iodine.

	Total Population	Women Supplementing	Women Not Supplementing	*p*
**Folate**	*n* = 66	*n* = 42	*n* = 24	
Dietary, µg	207.3 ± 96.4 (54.0–778.5)	221.5 ± 102.4 (100.3–778.5)	182.5 ± 81.0 (54.0–444.8)	0.114
Supplementary, µg	1511.4 ± 2246.8 (0.0–5400.0)	2375.0 ± 2429.4 (350.0–5400)		
Total, µg	1718.7 ± 2257.2 (54.0–5744.8)	2596.5 ± 2429.6 (512.0–5744.8)	182.5 ± 81.0 (54.0–444.8)	<0.001
Women meeting RNI, *n* (%)	43 (65.2)	42 (100)	1 (4.2)	
**Vitamin D**	*n* = 66	*n* = 37	*n* = 29	
Dietary, µg	1.7 ± 1.1 (0.1–5.0)	2.0 ± 1.1 (0.3–4.6)	1.3 ± 0.9 (0.1–3.5)	0.007
Supplementary, µg	5.8 ± 5.5 (0.0–25.0)	10.4 ± 2.5 (10.0–25.0)		
Total, µg	7.5 ± 5.9 (0.1–26.0)	12.4 ± 2.6 (10.3–26.0)	1.3 ± 0.9 (0.1–3.5)	<0.001
Women meeting RNI, *n* (%)	37 (56.1)	37 (100)	0 (0)	
**Iodine**	*n* = 66	*n* = 28	*n* = 38	
Dietary, µg	98.8 ± 54.4 (13.7–261.5)	107.4 ± 52.2 (39.9–217.2)	92.5 ± 55.9 (13.7–261.5)	0.082
Supplementary, µg	63.0 ± 74.0 (0.0–150.0)	148.6 ± 3.6 (140.0–150.0)		
Total, µg	161.8 ± 97.7 (13.7–367.2)	255.9 ± 52.2 (189.9–367.2)	92.5 ± 55.9 (13.7–261.5)	<0.001
Women meeting UK RNI, *n* (%)	34 (51.5)	28 (100)	6 (15.8)	
Women meeting EFSA RNI, *n* (%)	26 (39.4)	24 (85.7)	4 (14.3)	

Mean ± SD. *p* values for the difference between women taking supplements vs. those not taking supplements, independent *t* test.

**Table 3 nutrients-14-05135-t003:** Breakdown of supplements consumed by women in trimester 1.

	*n* (% of Total Population)
**5 mg folic acid**	**17 (26)**
+pregnancy multivitamin	12
+10 µg vitamin D/day	2
No additional supplement	3
**400 µg folic acid**	**24 (36)**
Within pregnancy multivitamin	19
+10 µg vitamin D/day	3
No additional supplement	2
**Vitamin D only**	**1**
25 µg/day	1

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
