# Peer review of "Dietary Intakes of Folate, Vitamin D and Iodine during the First Trimester of Pregnancy and the Association between Supplement Use and Demographic Characteristics amongst White Caucasian Women Living with Obesity in the UK"

_nutrients, 2022, doi:10.3390/nu14235135_

Round 1

Reviewer 1 Report

Thank you for asking me to review this interesting and well-written paper. I have some minor suggestions to make:

Abstract 

Indicate that the population of pregnant women are all in the UK.

Introduction

Another paper has been published recently exploring the micronutrient intake of pregnant women living with obesity (see Charnley, M., Newson, L., Weeks, A. and Abayomi, J., 2021. Pregnant women living with obesity: a cross-sectional observational study of dietary quality and pregnancy outcomes. Nutrients13(5), p.1652.).

It is also worth mentioning that the European Food Safety Authority (EFSA) recommends 200ug/day of iodine during pregnancy - so 140 ug/day may be insufficient. Also UK DRVs are for healthy people (not with a high BMI), so many RNIs for pregnant women with BMI > 30 may be inadequate.

It would be helpful to mention vitamin D recommendations from other countries here, as most European countries recommend higher then 10ug/day (usually 15-25 ug/day), even at more southerly latitudes (with better uv light exposure than the UK). So a RNI of 10ug/day may also be inadequate for pregnant women with BMI > 30, living in the UK.

Methods

clearly explained.

Results

Can you include ethnicity in table one - it is helpful to be aware that all were white Caucasian earlier in this section.

It would be useful to see how many women achieve the higher recommendations for iodine i.e. EFSA 200ug/day - not just UK recommendation.

Discussion

The recommendations for higher doses of folic acid and vitamin D for women with a BMI >30, have been around for a long time - so it is worrying that these messages are not being passed on to women. Also, many women may need to have higher doses of folic acid (5mg) prescribed by GPs as it is not available over the counter - so there is clearly a need to ensure that GPs and midwives are aware of the need for these higher doses too.

It would be helpful to have some further details regarding multi vitamin supplements commonly used in pregnancy here - i.e. which ones do and don't contain iodine? Also, which do and don't contain the higher doses of folic acid and/or vitamin D that are recommended for higher risk women? What about Healthy Start vitamins and the limitations of using these in women with a BMI > 30?

Limitations

Another limitation is possibly under reporting of dietary intake; this is well known in people with higher BMI (plus your energy intake data in Table one is rather low for an over weight population). Some discussion of this is needed here.

Conclusion

The need to inform health professionals (particularly GPs and midwives) regarding higher doses of supplements for higher risk women is also urgently required.

Author Response

Reviewer 1

Abstract

“Indicate that the population of pregnant women are all in the UK.”

This has been added to line 16.

Introduction

“Another paper has been published recently exploring the micronutrient intake of pregnant women living with obesity (see Charnley, M., Newson, L., Weeks, A. and Abayomi, J., 2021. Pregnant women living with obesity: a cross-sectional observational study of dietary quality and pregnancy outcomes. Nutrients, 13(5), p.1652.)”.

Thank you for pointing this interesting paper out. We have added some discussion of this paper to the introduction – lines 44-48.

‘It is also worth mentioning that the European Food Safety Authority (EFSA) recommends 200ug/day of iodine during pregnancy - so 140 ug/day may be insufficient. Also UK DRVs are for healthy people (not with a high BMI), so many RNIs for pregnant women with BMI > 30 may be inadequate.’

We have added some discussion of the EFSA recommendations to the introduction – lines 78-81.

‘It would be helpful to mention vitamin D recommendations from other countries here, as most European countries recommend higher then 10ug/day (usually 15-25 ug/day), even at more southerly latitudes (with better uv light exposure than the UK). So a RNI of 10ug/day may also be inadequate for pregnant women with BMI > 30, living in the UK.’

Discussion of these European variations has been added to lines 86-91.

Results

‘Can you include ethnicity in table one - it is helpful to be aware that all were white Caucasian earlier in this section’.

This has been added to Table 1.

‘It would be useful to see how many women achieve the higher recommendations for iodine i.e. EFSA 200ug/day - not just UK recommendation.’

We have added a row with this information to Table 2, and also described these trends in lines 168-170 and 172-173.

Discussion

‘The recommendations for higher doses of folic acid and vitamin D for women with a BMI >30, have been around for a long time - so it is worrying that these messages are not being passed on to women. Also, many women may need to have higher doses of folic acid (5mg) prescribed by GPs as it is not available over the counter - so there is clearly a need to ensure that GPs and midwives are aware of the need for these higher doses too.’

We have added a couple of sentences to make this important point in our conclusion – lines 366-369.

‘It would be helpful to have some further details regarding multi vitamin supplements commonly used in pregnancy here - i.e. which ones do and don't contain iodine? Also, which do and don't contain the higher doses of folic acid and/or vitamin D that are recommended for higher risk women? What about Healthy Start vitamins and the limitations of using these in women with a BMI > 30?’

We have added some discussion of this in lines 297-301 and also to our conclusion in lines 369-371.

Limitations

“Another limitation is possibly under reporting of dietary intake; this is well known in people with higher BMI (plus your energy intake data in Table one is rather low for an over weight population). Some discussion of this is needed here.”

We have added some discussion of this potential limitation to lines 342-347. We chose not to exclude women with low energy intakes, as when the researcher collected the 4 day diet diaries they were always discussed with the women to check for any missing data. Many women in their first trimester were still suffering with reduced appetite and pregnancy sickness, which was recorded and has been added to Table 1. 

Conclusion

“The need to inform health professionals (particularly GPs and midwives) regarding higher doses of supplements for higher risk women is also urgently required”.

Added to conclusion, lines 366-369.

Author Response

Reviewer 2

“Redfern and colleagues aim to investigate dietary and supplementary intakes of folate, vitamin D and iodine among a population of pregnant women with obesity and the association between supplement use and demographic characteristics. Although the topic is interesting, I have some concerns about the relevance of this article in the field of obese pregnant women nutrition because of its limitations, it would have been useful to know if the women enrolled in the study had previous miscarriages and if they were taking concomitant medications, the study is limited to the first trimester of pregnancy but it would have been interesting to know the percentage of women who took supplements until the end of pregnancy and the outcomes of supplementation/no supplementation.”

Unfortunately, women were not asked about previous miscarriages nor do we have information on the medication they may have been taking – we have added this to the potential limitations of the study – lines 338-342. We have now reported the changes in supplementation compliance across the three trimesters of pregnancy in Table 1 and describe these trends lines 148-152. Although we do have data on maternal and neonatal outcomes, we lack the statistical power to be able to analyse for differences between supplementers and non-supplementers across pregnancy.

“It should be specified in the title that the enrolled women were Caucasian.”

This has now been added to the title.

“Please add in the first section of Materials and Methods paragraph where and when was the study conducted.”

We have added details of the timeframe recruitment took place to the methods – line 114.

“Line 57 add a comma after “Ireland””

Added, now line 62.

“Line 88 please summarize the findings of the cited study or eliminate it. Anyway, why did you mention only one observational study but the references are two? Please correct it”.

We have 2 published papers from this work which is why we had added two citations. Rather than summarising the findings from these studies, which we feel would distract from the focus of this micronutrient work, we have removed these citations. Now line 101.

“Line 110 why did the researchers establish an observation period of only 4 days?”

We have added some information and justification for why we chose a 4 day dietary data collection period. Lines 125-129.

“Line 388 please eliminate the 0 before the first author’s surname.”

Reference list has been updated to reflect new citations and double checked for errors.

We appreciate your and the Reviewer’s time and insightful critique, which we believe has resulted in an improved manuscript. We believe we have addressed the Reviewer’s comments in sufficient detail and hope that our revised version is now suitable for publication in Nutrients. 

Round 2

Reviewer 2 Report

I believe that the manuscript has been sufficiently improved to warrant publication in Nutrients.